# Preoperative Factors for Lymphovascular Invasion in Prostate Cancer: A Systematic Review and Meta-Analysis

**DOI:** 10.3390/ijms25020856

**Published:** 2024-01-10

**Authors:** Jakub Karwacki, Marcel Stodolak, Łukasz Nowak, Paweł Kiełb, Wojciech Krajewski, Artur Lemiński, Tomasz Szydełko, Bartosz Małkiewicz

**Affiliations:** 1University Center of Excellence in Urology, Department of Minimally Invasive and Robotic Urology, Wroclaw Medical University, 50-556 Wroclaw, Poland; marcelstodolak@gmail.com (M.S.); lukasz.nowak@student.umw.edu.pl (Ł.N.); pawel.kielb@student.umw.edu.pl (P.K.); wojciech.krajewski@umw.edu.pl (W.K.); tomasz.szydelko@umw.edu.pl (T.S.); 2Department of Urology and Urological Oncology, Pomeranian Medical University, 70-111 Szczecin, Poland; artur.leminski@pum.edu.pl

**Keywords:** prostate cancer, lymphovascular invasion, radical prostatectomy, histopathological examination, prognostic factors, preoperative risk assessment, meta-analysis

## Abstract

Lymphovascular invasion (LVI) is one of the most important prognostic factors in prostate cancer (PCa) and is correlated with worse survival rates, biochemical recurrence (BCR), and lymph node metastasis (LNM). The ability to predict LVI preoperatively in PCa may be useful for proposing variations in the diagnosis and management strategies. We performed a systematic review and meta-analysis to identify preoperative clinicopathological factors that correlate with LVI in final histopathological specimens in PCa patients. Systematic literature searches of PubMed, Embase, and Web of Science were performed up to 31 January 2023. A total of thirty-nine studies including 389,918 patients were included, most of which were retrospective and single-center. PSA level, clinical T stage, and biopsy Gleason score were significantly correlated with LVI in PCa specimens. Meta-analyses revealed that these factors were the strongest predictors of LVI in PCa patients. Prostate volume, BMI, and age were not significant predictors of LVI. A multitude of preoperative factors correlate with LVI in final histopathology. Meta-analyses confirmed correlation of LVI in final histopathology with higher preoperative PSA, clinical T stage, and biopsy Gleason score. This study implies advancements in risk stratification and enhanced clinical decision-making, and it underscores the importance of future research dedicated to validation and exploration of contemporary risk factors in PCa.

## 1. Introduction

Prostate cancer (PCa) is a prevalent malignancy with substantial clinical implications, variable outcomes, and an incidence of 1.4 million new cases in 2020 worldwide [1]. Lymphovascular invasion (LVI), a histopathological finding commonly defined as the unequivocal presence of tumor cells within endothelium lined spaces, has been associated with aggressive disease features and adverse prognostic outcomes, including lymph node invasion, and decreased survival rates [2,3,4]. Accurate preoperative prediction of LVI may be crucial for treatment decision-making in PCa patients. Additionally, both LVI and associated factors might provide added value to existing nomograms for nodal involvement [5,6,7,8] and risk classification systems [9].

The objective of this review was to identify clinicopathological factors that can predict LVI in PCa patients prior to undergoing radical prostatectomy (RP). By synthesizing the available evidence, we attempt to provide a comprehensive understanding of the predictive value of preoperative factors for LVI in PCa, and thus, for tumor aggressiveness and disease progression. The findings of this study may contribute to improved risk stratification by facilitating the identification of those at higher risk of LVI and its associated adverse outcomes.

## 2. Results

### 2.1. Study Characteristics

We identified thirty-nine eligible studies, including 389,997 patients, 33,995 of whom were LVI-positive (8.7%) (Table 1). One study [10] included 79 PCa patients, although the exact number of LVI-positive patients was not disclosed. Of thirty-nine included papers, seventeen were retrospective and single-center [11,12,13,14,15,16,17,18,19,20,21,22,23,24,25,26,27], twelve were retrospective and multi-center [28,29,30,31,32,33,34,35,36,37,38,39], eight were prospective and single-center [10,40,41,42,43,44,45,46], and two were retrospective and database-registry-based [47,48]. Most commonly, LVI was defined as the unequivocal presence of tumor cells within endothelial-lined spaces. Figure 1 displays a PRISMA flowchart depicting the inclusion of articles. 

The initial database search yielded 1073 records, and after removing 439 duplicates, 634 articles underwent abstract analysis. Of these, 234 were excluded based on article type, and an additional 6 were excluded due to non-English language. The screening process for study identification excluded 288 studies deemed ‘not relevant to this review’, primarily involving malignancies other than PCa or animal samples. A total of 106 articles were retrieved and assessed for eligibility through comprehensive manuscript analysis. Ultimately, 39 articles met the inclusion criteria and were included in the systematic review.

The NOS scores for the included studies varied from 4 to 9. Consequently, the methodological quality of the eligible manuscripts was categorized as moderate or high (see Appendix A) and thus deemed appropriate for this systematic review and meta-analysis. It is noteworthy that all selected papers exhibited a high RoB, primarily attributed to their retrospective design. The RoB assessment, generated using the robvis tool [49], is presented in Figure 2.

### 2.2. Meta-Analyses

Meta-analyses concerned six preoperative clinicopathological factors: age, BMI, clinical T stage, biopsy Gleason score (GS), preoperative PSA, and prostate volume (Figure 3 and Figure 4). Factors associated with occurrence of LVI in RP specimens included clinical T3 stage (*p* < 0.0001; OR = 3.54), biopsy Gleason score ≥8 (*p* < 0.00001; OR = 4.17), and preoperative PSA (*p* = 0.0008; MD = 5.53). Age (*p* = 0.10; MD = 0.64), BMI (*p* = 0.88; MD = −0.03), and prostate volume (*p* = 0.71; MD = −0.85) were uncorrelated with LVI.

#### 2.2.1. Age

A total of twenty-one studies investigated the correlation between age and LVI [11,12,13,17,19,21,22,23,24,25,26,28,30,32,36,37,38,39,46,47,48]. Due to substantial diversity of data presentation, eleven of them were excluded, leaving ten studies that made the final meta-analysis [11,13,17,23,24,25,28,30,37,39]. A total of 4107 patients were analyzed, 588 of whom were LVI-positive. Overall, results from the random-effects meta-analysis model indicate that older age is not a risk factor for LVI in the final histopathology (*p* = 0.10; MD = 0.64; 95% CI −0.12, 1.39).

#### 2.2.2. BMI

A total of five studies investigated the correlation between BMI and LVI [17,25,26,27,39]. Due to substantial diversity of data presentation, one of them was excluded, leaving four studies that made the final meta-analysis [17,25,26,39]. A total of 3999 patients were analyzed, 592 of whom were LVI-positive. Overall, results from the random-effects meta-analysis model indicate that BMI is uncorrelated with LVI in the final histopathology (*p* = 0.88; MD = −0.03; 95% CI 0.40, 0.34).

#### 2.2.3. Clinical T Stage 

A total of eight studies investigated the correlation between clinical T stage and LVI [11,19,24,26,30,31,37,38]. Due to substantial diversity of data presentation, three of them were excluded, leaving five studies that made the final meta-analysis [11,24,30,37,38]. A total of 2866 patients were analyzed, 283 of whom were LVI-positive. Overall, results from the random-effects meta-analysis model indicate that stage T3 is associated with LVI in the final histopathology (*p* < 0.0001; OR = 3.54; 95% CI 1.97, 6.38). Stage T1 has a significantly lower probability of LVI (*p* = 0.02; OR = 0.57; 95% CI 0.36, 0.91). Clinical stage T2 did not correlate with LVI (*p* = 0.21; OR = 1.49; 95% CI 0.80, 2.79).

#### 2.2.4. Gleason Score

A total of eight studies investigated the correlation between biopsy GS and LVI [11,19,20,26,30,31,38,39]. Due to substantial diversity of data presentation, four of them were excluded, leaving four studies that made the final meta-analysis [11,26,30,39]. A total of 4299 patients were analyzed, 591 of whom were LVI-positive. Overall, results from the random-effects meta-analysis model indicate that GS ≥8 is correlated with LVI in the final histopathology (*p* < 0.00001; OR = 4.17; 95% CI 2.38, 7.31). Patients with a GS of 6 have a statistically lower probability of LVI in the final specimens (*p* < 0.00001; OR = 0.28; 95% CI 0.16, 0.49). A GS of 7 did not correlate with LVI (*p* = 0.33; OR = 1.27; 95% CI 0.79, 2.07).

#### 2.2.5. Preoperative PSA 

A total of twenty-seven studies investigated the correlation between preoperative PSA levels and LVI [11,12,13,17,19,20,21,22,23,24,25,26,28,30,31,32,34,37,38,39,42,43,44,45,46,47,48]. Due to substantial diversity of data presentation, eighteen of them were excluded, leaving nine studies that made the final meta-analysis [19,23,24,25,28,30,37,39,45]. A total of 3607 patients were analyzed, 516 of whom were LVI-positive. Overall, results from the random-effects meta-analysis model indicate that high preoperative PSA is a risk factor for LVI in the final histopathology (*p* = 0.0008; MD = 5.53; 95% CI 2.30, 8.75).

#### 2.2.6. Prostate Volume

A total of four studies investigated the correlation between prostate volume and LVI [24,25,26,39]. Due to substantial diversity of data presentation, one of them was excluded, leaving three studies that made the final meta-analysis [24,25,39]. A total of 1854 patients were analyzed, 328 of whom were LVI-positive. Overall, results from the random-effects meta-analysis model indicate that prostate volume is uncorrelated with LVI in final specimens (*p* = 0.71; MD = −0.85; 95% CI −5.33, 3.63).

### 2.3. Results of the Systematic Review

The review of collected data revealed several clinical, pathological, imaging, and other prognostic factors correlated with LVI in RP specimens (Table 2 and Figure 5).

#### 2.3.1. Preoperative Clinical Factors

We identified ten preoperative clinical factors, acquired from blood or urine samples, correlated with LVI: three of them were associated with PSA—preoperative PSA (including PSA at diagnosis) was reported by twenty-seven studies [11,12,13,17,19,20,21,22,23,24,25,26,28,30,31,32,34,37,38,39,42,43,44,45,46,47,48], and nineteen of them revealed a significant correlation [11,12,13,17,19,21,22,23,24,26,30,34,37,38,39,42,46,47,48]; PSA density (PSAD) was reported by two studies [24,37]; and percent free PSA was reported by one study [44]. The other factors significantly correlated with LVI included carrying at least one G allele at carbon anhydrase 9 (CA9) rs3829078 (acquired from serum) [42], serum epidermal growth factor receptor (EGFR) [43], urine In1-ghrelin [10], serum prostate stem cell antigen (PSCA) messenger ribonucleic acid (mRNA) expression [41], red blood cell distribution width (RDW-SD) [17], serum cholinesterase [29], low serum total cholesterol [14], serum urokinase-type plasminogen activator (uPA) [31], and serum uPA receptor (uPAR) [31]. Other, uncorrelated preoperative clinical factors included PSA velocity [34] and platelets [18].

#### 2.3.2. Preoperative Pathological Factors

This systematic review revealed four preoperative pathological factors correlated with LVI: biopsy GS, reported by eight studies [11,19,20,26,30,31,38,39] and associated with LVI by four of them [26,30,38,39]; percent of positive biopsy cores (PPBC), reported by four studies [11,30,35,37] and significantly correlated with LVI by three of them [11,30,37]; number of positive biopsy cores, reported by one study [39]; and disialosyl globopentaosyl ceramide (DSGb5) expression in biopsy specimens, reported by one study [15]. Other pathological factors in our review were uncorrelated in particular studies, such as perineural invasion in biopsy specimens [20,40] and total number of biopsy cores [39].

#### 2.3.3. Preoperative Imaging-Related Factors

Our review identified three preoperative imaging-related factors associated with LVI: PIRADS score, reported and correlated with LVI by one study [33]; prostate volume, reported by four studies [24,25,26,39] and correlated with LVI by one of them [39]; and tumor volume, associated with LVI by one study [38].

#### 2.3.4. Other Preoperative Factors

We identified various other preoperative factors correlated with LVI, which included the following: age, reported by twenty studies [11,12,13,17,21,22,23,24,25,26,28,30,32,36,37,38,39,46,47,48] and correlated by four of them [25,38,47,48] (two studies did not provide the *p*-value [21,46], although one of them [21] reported no correlation); body mass index (BMI), reported by five studies [17,25,26,27,39] and correlated with LVI by one of them [27]; digital rectal examination abnormality, reported by two studies [20,26] and associated with LVI by one of them [26]; race, reported by two studies [23,48] and associated with LVI by one of them [48] (in case of the study by Jamil et al. [48], African American (AA) and other races were associated with higher LVI prevalence in comparison to Caucasian race); and clinical tumor stage (cT), reported by eight studies [11,19,24,26,30,31,37,38] and associated with LVI by four of them [11,26,30,38]. Additionally, one study [13] reported a correlation between the risk group classification and LVI. Another study [25] revealed that D’Amico classification correlated with LVI. Both classification systems are multifactorial; thus, we did not include them in Figure 5.

## 3. Discussion

LVI is considered an adverse pathological feature and has been consistently associated with an increased risk of disease progression and poor outcomes. Studies have shown that patients with LVI are more likely to have biochemical recurrence, tumor metastasis, and lower survival rates [2,3,4]. Additionally, LVI is also associated with unfavorable sub-pathologies, including cribriform and intraductal patterns [50]. In this study, we aimed to identify preoperative clinical and pathological factors that correlate with LVI, to enhance our ability to predict its occurrence and guide treatment decisions.

Predicting the occurrence of LVI in prostate cancer may have significant implications for patient prognosis and treatment planning. By preoperatively identifying patients at higher risk of LVI, clinicians could potentially offer closer surveillance, neoadjuvant therapy, or more aggressive surgical strategies, for instance, omitting the nerve-sparing technique, opting for wider tissue margins, or using a more extended lymph node dissection (LND) template.

Accurate detection of metastatic lymph nodes enables clinicians to make informed decisions regarding appropriate treatment approaches, such as extended LND or targeted radiation therapy [51]. Traditional imaging techniques such as computed tomography (CT) and magnetic resonance imaging (MRI) have limitations in detecting small or micrometastatic lymph nodes [52]. Thus, incorporating the evaluation of LVI status into preoperative assessments can help identify patients who may benefit from more extended LND templates or adjuvant therapies. Despite the established association between LVI and lymph node metastasis (LNM), it is noteworthy that existing prognostic nomograms and other risk evaluation tools often do not incorporate LVI and many of its predictors as risk factors. To improve risk stratification and enhance the clinical utility of prognostic tools, it might be beneficial to incorporate LVI and associated risk factors into algorithms to provide a more comprehensive assessment of the individual patient’s risk profile, facilitating more informed treatment decisions. Furthermore, identified prognostic factors for LVI could have an impact on the biopsy-associated decision process. Many pathologists do not include LVI in biopsy specimen reports, which may be caused by the relative rarity of this finding [53] or the resemblance to many histopathological artifacts [50]. Nevertheless, the EAU guidelines state that patients with LVI-positive biopsy should be excluded from active surveillance [9]. Thus, including additional markers could have an impact on clinicians’ and pathologists’ decisions regarding inclusion of LVI in the biopsy report.

Predicting LVI occurrence enables researchers to identify high-risk patient populations for further investigation. Clinical trials can be designed specifically to evaluate novel treatment approaches or interventions targeting patients with LVI. Improved understanding of LVI’s biological mechanisms and associated biomarkers may lead to the development of targeted therapies or interventions as well as more accurate prognostic tools. 

It is crucial to acknowledge that further research and data are needed to refine the accuracy of LVI prediction. While certain clinicopathological factors, such as preoperative PSA, cT stage, and biopsy GS, have shown significant correlations with LVI, there may be additional as-yet-unknown factors that are associated with its occurrence. Therefore, ongoing investigations and prospective studies are warranted to enhance our understanding of LVI predictors and develop comprehensive models that encompass a broader range of variables.

An intriguing aspect of this study is the observed lack of correlation between age and LVI. While our findings suggest no significant association, it is noteworthy that existing literature in the field presents inconsistent perspectives on the role of age as a risk factor in PCa [54,55]. Some studies indicate an association between older age and worse prognosis with unfavorable outcomes, whereas others underscore the insignificance of this correlation [56]. Importantly, the studies included in our meta-analysis on the age–LVI association were primarily retrospective, contributing to the complexity of our observations. Furthermore, it is crucial to highlight that although the heterogeneity in our meta-analysis was low (I^2^ = 25%), the highest-volume studies by Rakic et al. and Jamil et al. (see Table 1), unfortunately, were not incorporated into the meta-analysis due to differences in data presentation. These studies, displaying a statistically significant correlation between LVI and age, were excluded due to the unavailability of specific parameters (mean, range, SD) necessary for the meta-analysis. This underscores the challenge of synthesizing evidence from diverse studies and emphasizes the need for standardized reporting in future research.

Our study is burdened with some limitations that should be acknowledged. Firstly, many included studies in our systematic review were of retrospective nature, which introduces inherent limitations such as potential selection bias and the inability to establish causal relationships. Additionally, the reliance on retrospective data may be associated with incomplete or missing information, leading to potential inaccuracies or inconsistencies in the results. Secondly, the studies included in our analysis were predominantly single-center studies, which may limit the generalizability of the findings. The results might have been influenced by specific patient populations, treatment protocols, or institutional practices. Thirdly, the factors associated with LVI in our meta-analyses are major prognostic factors for PCa aggressiveness itself. To establish the mechanisms behind LVI, it would be valuable to search for prognostic factors correlated with LVI, but not with a higher stage, grade, or other adverse pathological outcomes, in future studies.

In conclusion, our study has identified several preoperative clinical and pathological factors that correlate with LVI in PCa patients. These factors hold potential as predictive markers for the presence of LVI and may improve risk stratification and treatment planning. The association of LVI with worse outcomes, as demonstrated in previous studies, underscores the importance of predicting LVI to guide treatment decisions. Further research is warranted to validate our findings, investigate underlying mechanisms, and develop comprehensive risk stratification models that incorporate LVI as a predictive factor.

## 4. Materials and Methods

### 4.1. Search Strategy

Two review authors (J.K. and M.S.) independently performed a computerized systematic literature search of the PubMed, Embase, and Web of Science databases in June 2023. Only articles written in English with the full text available, without time limitations, were considered. The following terms and keywords were employed:PubMed: ‘prostate cancer’ AND (‘microvascular invasion’ or ‘lymphovascular invasion’) using Medical Subject Headings (MeSH) terms;Embase: (‘prostate cancer’/exp OR ‘prostate cancer’ OR ((‘prostate’/exp OR prostate) AND (‘cancer’/exp OR cancer))) AND (lymphovascular OR microvascular) AND invasion;Web of Science: (ALL = (prostate cancer)) AND ALL = (lymphovascular invasion OR microvascular invasion).

The references of the relevant review articles were also manually screened to ensure that no additional eligible papers were inadvertently omitted.

### 4.2. Inclusion Criteria

All authors participated in the design of the search strategy and inclusion criteria. Our procedure for evaluating records identified during the literature search followed the Preferred Reporting Items for Systematic Reviews and Meta-analyses (PRISMA) criteria [57]. The study protocol was registered a priori on the International Prospective Register of Systematic Reviews (PROSPERO) with the registration number CRD42023389021. The final list of included articles was selected with the consensus of all collaborating authors, verifying that they met the inclusion criteria.

### 4.3. Study Eligibility and Quality/Risk of Bias Assessment

Studies were assessed for eligibility using the PICO (population, intervention, comparison, outcome) approach:Population: patients with PCa who underwent RP.Intervention: RP and final histopathological examination; patients with LVI in final specimens.Comparison: patients without LVI in final histopathology.Outcome: preoperative clinicopathological factors and their correlation with LVI in RP specimens.

Additionally, studies included in the systematic review were required to fulfill the following inclusion criteria: (1) original article, (2) human research, (3) English language, (4) access to the full manuscript, (5) prostate cancer, (6) LVI evaluated in RP specimens, and (7) association between various preoperative clinicopathological factors and LVI assessed (*p*-value revealed). The exclusion criteria were: (1) non-comparative studies—reviews, letters, conference papers, editorial comments, replies from authors, or case reports, (2) studies not reporting outcomes of interest. 

The included studies underwent quality evaluation based on the Newcastle–Ottawa Scale (NOS) guidelines [58], encompassing three key domains: (1)Selection of the study population;(2)Comparability of the groups;(3)Ascertainment of the outcome.

A summary of the quality assessment is provided in Appendix A.

The assessment of the risk of bias (RoB) followed the principles outlined in the *Cochrane Handbook for Systematic Reviews of Interventions* [59].

### 4.4. Statistical Analysis

Means and standard deviations (SDs) or medians and interquartile ranges (IQRs) were utilized for continuous variables. All median and IQR values were converted to means and SDs using the methodology described by Hozo et al. [60]. Pooled estimates were obtained using means and SDs for continuous variables and event rates for categorical variables. The effect measure used for continuous variables was the mean difference (MD), and odds ratio (OR) was used for categorical variables. The study applied 95% confidence intervals (95% CI). Heterogeneity was assessed through the Chi-square-based Q test and I^2^. While not every meta-analysis demonstrated significant heterogeneity among the included studies, random-effect models were consistently employed in each case. This decision was based on the high overall heterogeneity observed across all studies included in the systematic review.

It is noteworthy to highlight that specific studies were excluded from certain meta-analyses due to variations in data presentation. For instance, when median and range information were available, we could derive the mean and incorporate the corresponding articles into the meta-analysis. However, in instances where articles solely provided the median value for a particular parameter, inclusion in the meta-analysis became unfeasible. Similar criteria were applied to categorical parameters (e.g., cT stage). If the data presentation did not permit the event rates to be inferred (e.g., number of patients with cT1, cT2, and cT3, with and without LVI), the study was excluded from the meta-analysis, even if it presented an association with a certain parameter. In the subsections of each meta-analysis, all included and excluded studies were meticulously cited, enabling a comprehensive individual examination of each excluded study.

## 5. Conclusions

In conclusion, our study demonstrates that several preoperative clinicopathological factors are significantly correlated with LVI in final histopathological specimens of PCa patients. Notably, preoperative PSA levels, clinical T stage, and biopsy Gleason score emerged as the strongest predictors of LVI in PCa patients. These findings highlight the importance of considering these factors in the preoperative assessment of PCa patients to accurately predict the likelihood of LVI occurrence. While factors such as prostate volume, BMI, and age did not exhibit significant correlations with LVI, further research is warranted to explore additional potential predictors and refine the predictive models. Additionally, integrating novel predictors into future approaches, such as the development of preoperative nomograms, could enhance risk stratification, disease staging, and treatment decision-making. Overall, our findings contribute to a better understanding of LVI in PCa and provide valuable insights for clinical practice and future research endeavors.

## Figures and Tables

**Figure 1 ijms-25-00856-f001:**
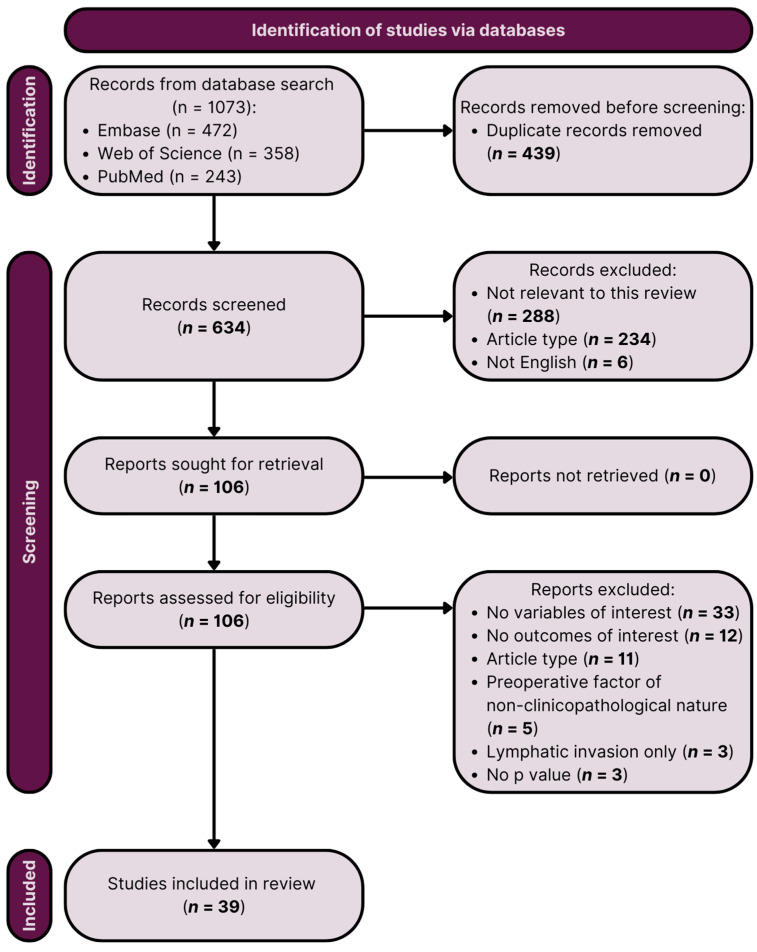
PRISMA (Preferred Reporting Items of Systematic Reviews and Meta-Analyses) flow diagram of the study selection process.

**Figure 2 ijms-25-00856-f002:**
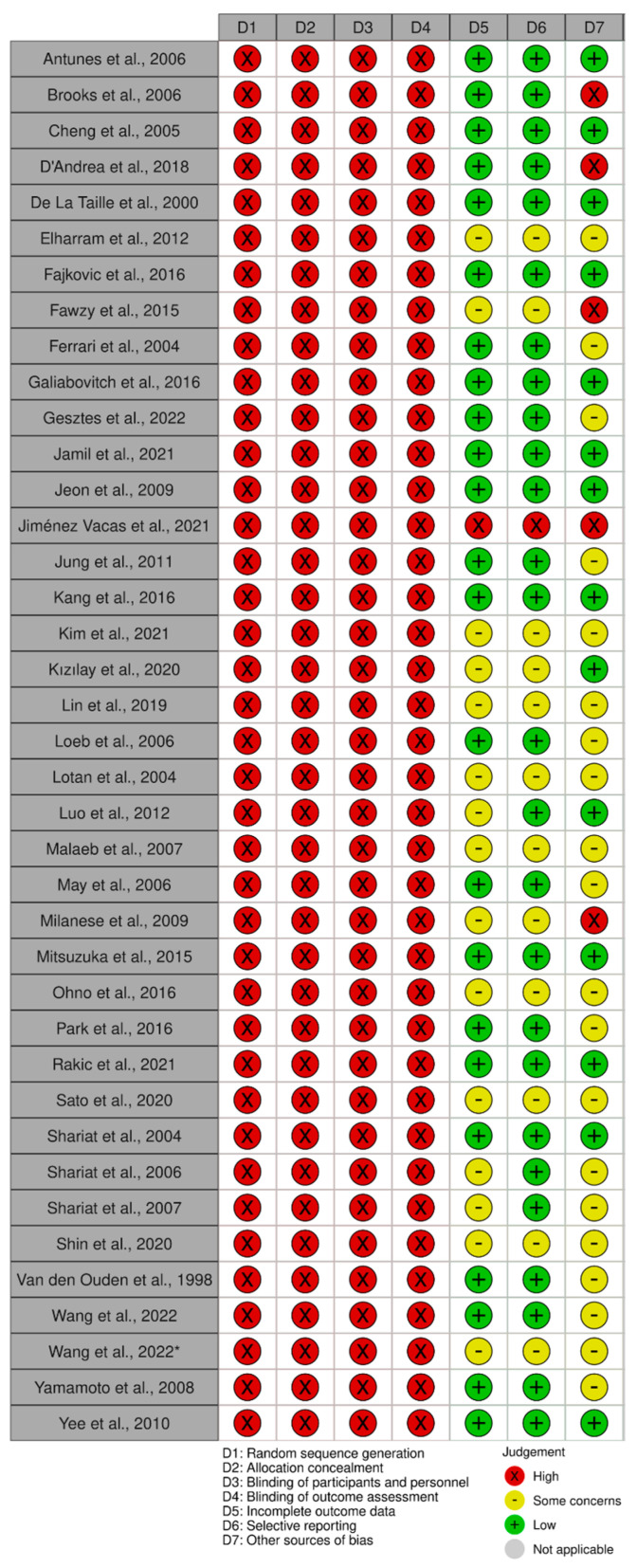
Risk of bias assessment chart [10,11,12,13,14,15,16,17,18,19,20,21,22,23,24,25,26,27,28,29,30,31,32,33,34,35,36,37,38,39,40,41,42,43,44,45,46,47,48]. * [18].

**Figure 3 ijms-25-00856-f003:**
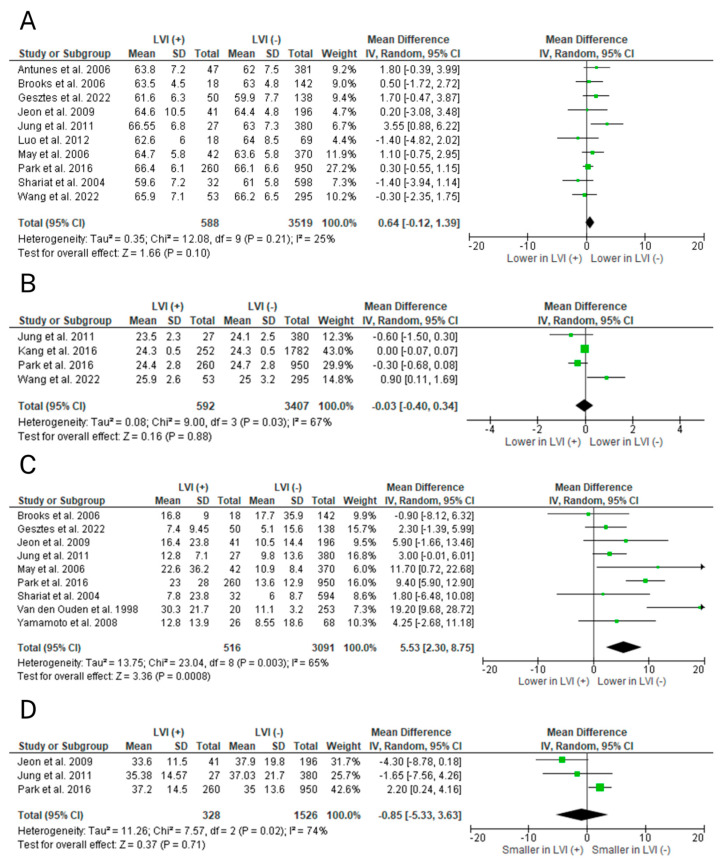
Forest plots of mean differences (MD) in random effects models predicting lymphovascular invasion (LVI) preoperatively: (**A**) age, (**B**) body mass index (BMI), (**C**) prostate-specific antigen (PSA), and (**D**) prostate volume [11,13,17,19,23,24,25,26,28,30,37,39,45]. CI = confidence interval; df = degree of freedom; IV = inverse variance; M-H = Mantel–Haenszel; SD = standard deviation.

**Figure 4 ijms-25-00856-f004:**
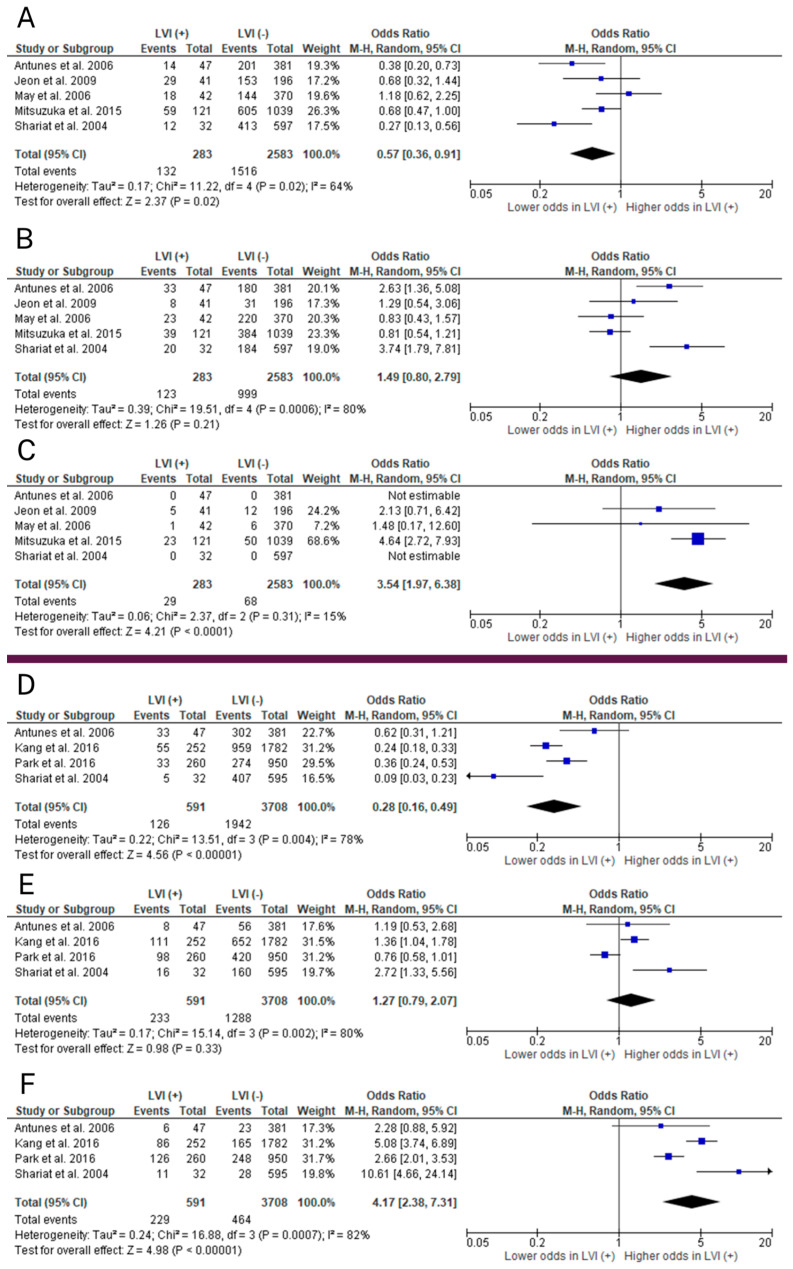
Forest plots of odds ratios (OR) in random effects models predicting lymphovascular invasion (LVI) preoperatively: clinical T stage (cT)—(**A**) cT1, (**B**) cT2, and (**C**) cT3; biopsy Gleason score (GS)—(**D**) GS = 6, (**E**) GS = 7, and (**F**) GS ≥ 8 [11,24,26,30,37,38,39]. CI = confidence interval; df = degree of freedom; IV = inverse variance; M-H = Mantel–Haenszel; SD = standard deviation.

**Figure 5 ijms-25-00856-f005:**
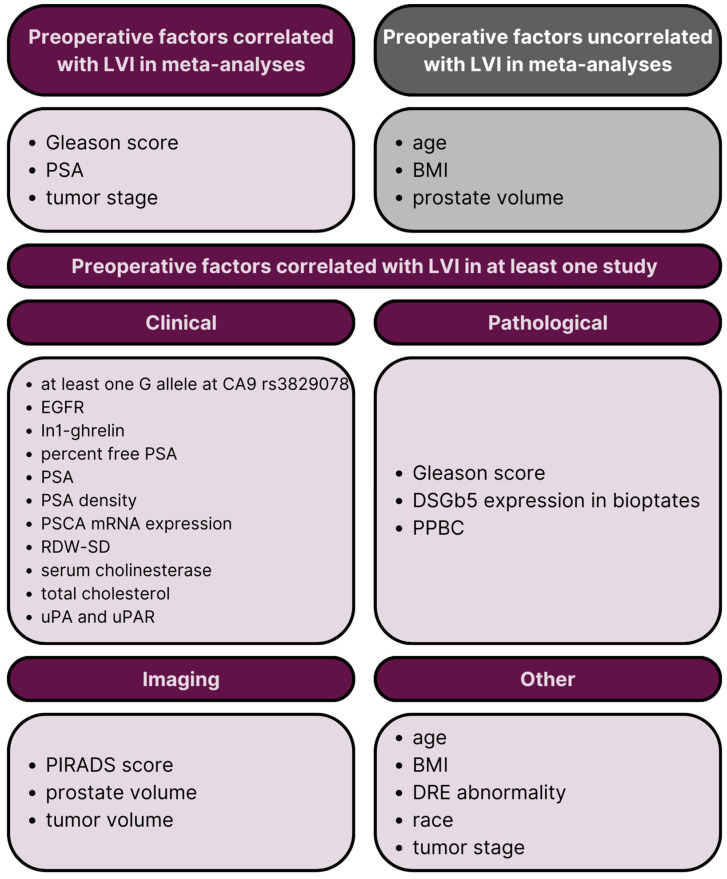
A visual overview of various clinicopathological preoperative factors that correlated with LVI in at least one study. LVI = lymphovascular invasion; PSA = prostate-specific antigen; BMI = body mass index; CA9 = carbon anhydrase 9; EGFR = epidermal growth factor receptor; PSCA = prostate stem cell antigen; mRNA = messenger ribonucleic acid; RDW-SD = red blood cell distribution width; uPA = urokinase-type plasminogen activator; uPAR = urokinase-type plasminogen activator receptor; DSGb5 = disialosyl globopentaosyl ceramide; PPBC = percent of positive biopsy cores; PIRADS = Prostate Imaging Reporting & Data System; DRE = digital rectal examination.

**Table 1 ijms-25-00856-t001:** Study characteristics of the 39 included studies.

Study	Country	Journal	Study Design	Study Period	LVI+ and LVI− [*n*]	LVI+ [*n*]	LVI+ [%]	LVI− [*n*]	Definition of LVI	Preoperative Predictors of LVI (*p*-Value)	NOS
Antunes et al., 2006 [11]	Brazil	*International Brazilian Journal of Urology*	retrospective, single-center	1993–2000	428	47	11.0	381	presence of tumor cells within an endothelium lined space	age (0.372) cT stage (0.003) GS (0.169) PPBC (mean: <0.001; ranges: 0.006) PSA (mean: 0.004; ranges: 0.006)	8
Brooks et al., 2006 [28]	USA	*Cancer*	retrospective, multi-center	1991–2001	160	18	11.3	142	unequivocal presence of tumor cells within a vascular or lymphatic, endothelial-lined space	age (0.60) PSA (0.44)	7
Cheng et al., 2005 [12]	USA	*The Journal of Urology*	retrospective, single-center	1990–1998	504	106	21.0	398	unequivocal presence of tumor cells in an endothelium lined space	age (0.20) PSA (<0.0001)	8
D’Andrea et al., 2018 [29]	multi-center	*Urologic Oncology*	retrospective, multi-center	2000–2011	6041	693	11.5	5348	unequivocal presence of tumor cells within an endothelium lined space without underlying muscular walls	serum cholinesterase (<0.01)	7
De La Taille et al., 2000 [20]	USA	*European Urology*	retrospective, single-center	1993–1998	241	30	12.4	211	unequivocal presence of tumor cells within simple endothelial-lined tissue spaces without underlying muscular walls	DRE abnormalities (0.16) GS (0.14) PNI (0.09) PSA (0.09)	8
Elharram et al., 2012 [40]	Canada	*The Canadian Journal of Urology*	prospective, single-center	2008–2010	138	5	3.6	133	none	PNI (0.499)	5
Fajkovic et al., 2016 [32]	multi-center	*Urologic Oncology*	retrospective, multi-center	2000–2011	6678	767	11.5	5911	presence of tumor cells within an endothelium lined space	age (0.054) PSA (0.59)	8
Fawzy et al., 2015 [41]	Egypt	*Medical Oncology*	prospective, single-center	not specified	112	52	46.4	60	none	PSCA mRNA (0.02)	6
Ferrari et al., 2004 [21]	USA	*Adult Urology*	retrospective, single-center	1984–1999	620	110	17.7	510	LVI was in small channels consisting of only an endothelial layer, identified by its smooth luminal surface, and lined with one or two narrow endothelial cells	age (NA) PSA (<0.0001)	8
Galiabovitch et al., 2016 [22]	Australia	*BJUI International*	retrospective, single-center	2004–2012	1267	82	6.5	1185	unequivocal presence of tumor cells in an endothelial-lined space	age (0.26) PSA (<0.001)	8
Gesztes et al., 2022 [23]	USA	*Scientific Reports*	retrospective, single-center	1993–2013	188	50	26.6	138	tumor cells’ presence within spaces lined by lymphovascular endothelium with characteristic podoplanin staining	age (0.14) PSA (0.0027) race (0.0796)	8
Jamil et al., 2021 [48]	multi-center	*Clinical Genitourinary Cancer*	retrospective, database registry	2010–2015	232,704	17,758	7.6	214,946	presence of tumor cells in lymphatic channels or blood vessels within the primary tumor	age (<0.0001) PSA (0.01) race (AA: <0.0001; other: 0.002)	8
Jeon et al., 2009 [24]	Korea	*International Journal of Urology*	retrospective, single-center	1995–2004	237	41	17.3	196	unequivocal presence of tumor cells within an endothelium lined space without muscular walls	age (0.865) PSA (0.002) prostate volume (0.160) PSA density (<0.001) cT stage (0.544)	8
Jiménez Vacas et al., 2021 [10]	Spain	*The Journal of Clinical Endocrinology & Metabolism*	prospective, single-center	2013–2015	79	not specified	-	not specified	none	In1-ghrelin serum level (<0.001)	4
Jung et al., 2011 [25]	Korea	*Annals of Surgical Oncology*	retrospective, single-center	2005–2009	407	27	6.6	380	unequivocal presence of tumor cells in an endothelium lined space	age (0.024) BMI (0.268) D'Amico classification (<0.001) PSA (0.075) prostate volume (0.647)	8
Kang et al., 2016 [26]	Korea	*Annals of Surgical Oncology*	retrospective, single-center	2003–2014	2034	252	12.4	1782	presence of tumor emboli in intraprostatic vessels, particularly within the lumen of the endothelium	age (0.303) GS (<0.001) BMI (0.857) cT stage (<0.001) DM (0.192) DRE abnormalities (<0.001) hypertension (0.117) prostate volume (0.148) PSA (<0.001)	8
Kim et al., 2021 [27]	Korea	*International Journal of Clinical Oncology*	retrospective, single-center	1997–2019	389	59	15.2	330	none	BMI (0.043)	6
Kızılay et al., 2020 [33]	Turkey	*Prostate International*	retrospective, multi-center	not specified	177	10	5.6	167	none	PIRADS score (0.032)	5
Lin et al., 2019 [42]	Taiwan	*Urologic Oncology*	prospective, single-center	2012–2017	579	97	16.8	482	none	carrying at least one G allele at CA9 rs3829078 (0.008) PSA (<0.001)	7
Loeb et al., 2006 [34]	USA	*Urology*	retrospective, multi-center	1989–2004	1709	118	6.9	1591	presence of tumor emboli in small intraprostatic vessels	PSA (<0.0001) PSA velocity (0.3)	8
Lotan et al., 2004 [35]	USA	*The Journal of Urology*	retrospective, multi-center	1994–2002	605	32	5.3	573	none	PPBC (0.736)	6
Luo et al., 2012 [13]	Taiwan	*Kaohsiung Journal of Medical Sciences*	retrospective, single-center	1998–2010	87	18	20.7	69	none	age (0.746) PSA (0.009) risk classification (0.003)	6
Malaeb et al., 2007 [36]	USA	*Urologic Oncology*	retrospective, multi-center	1994–2002	628	34	5.4	594	none	age (0.559)	6
May et al., 2006 [37]	Germany	*BJUI International*	retrospective, multi-center	1996–2003	412	42	10.2	370	unequivocal existence of tumor cells in an endothelium lined space with no underlying muscular walls	age (0.261) cT stage (0.811) PSA density (<0.001) PSA (<0.001) PPBC (0.001)	8
Milanese et al., 2009 [43]	Italy	*The Journal of Urology*	prospective, single-center	2005–2006	30	4	13.3	26	none	EGFR (0.005) PSA (0.120) uPAR (0.094)	6
Mitsuzuka et al., 2015 [38]	Japan	*Prostate Cancer and Prostatic Disease*	retrospective, multi-center	2000–2009	1160	121	10.4	1039	unequivocal presence of tumor cells within endothelial-lined channels on routine light microscopic examination	age (0.017) GS (<0.001) cT stage (<0.001) PSA (0.006) tumor volume (<0.001)	8
Ohno et al., 2016 [14]	Japan	*Molecular and Clinical Oncology*	retrospective, single-center	2002–2010	562	148	26.3	414	none	low serum total cholesterol level (0.014)	6
Park et al., 2016 [39]	South Korea	*Scientific Reports*	retrospective, multi-center	2001–2012	1210	260	21.5	950	presence of cancer cells within an arterial, venous, or lymphatic lumen on routine hematoxylin and eosin sections	age (0.603) GS (<0.001) BMI (0.213) no. of positive biopsy cores (<0.001) PSA (<0.001) prostate volume (0.025) total no. of biopsy cores (0.088)	8
Rakic et al., 2021 [47]	USA	*Urologic Oncology*	retrospective, database registry	2010–2015	126,682	12,632	10.0	114,050	presence of tumor cells in lymphatic channels or blood vessels within the primary tumor, but not the lymph nodes	age (<0.0001) PSA (<0.0001)	8
Sato et al., 2020 [15]	Japan	*The Tohoku Journal of Experimental Medicine*	retrospective, single-center	2005–2011	116	12	10.3	104	none	DSGb5 expression in prostate biopsy specimens (0.027)	5
Shariat et al., 2004 [30]	USA	*The Journal of Urology*	retrospective, multi-center	1994–2004	630	32	5.1	598	unequivocal presence of tumor cells within an endothelium lined space without underlying muscular walls	age (0.1) cT stage (<0.001) GS (<0.001) PPBC (<0.001) PSA (0.004)	8
Shariat et al., 2006 [44]	USA	*European Urology*	prospective, single-center	1994–2002	351	13	3.7	338	none	% fPSA (0.003) PSA (0.901)	6
Shariat et al., 2007 [31]	USA	*Journal of Clinical Oncology*	retrospective, multi-center	1994–2004	255	29	11.4	226	none	cT stage (0.078) GS (0.47) PSA (0.238) uPA (0.002) uPAR (<0.001)	7
Shin et al., 2020 [16]	Korea	*Prostate International*	retrospective, single-center	2009–2016	216	14	6.5	202	none	visible tumor in MRI (0.876)	5
Van den Ouden et al., 1998 [45]	The Netherlands	*Urologia Internationalis*	prospective, single-center	1977–1994	273	20	7.3	253	unequivocal presence of tumor cells within endothelial-lined spaces	PSA (0.1)	7
Wang et al., 2022 [17]	China	*Frontiers in Endocrinology*	retrospective, single-center	2018–2021	348	53	15.2	295	presence of tumor cells within an endothelial-lined space that is usually devoid of a muscular wall	age (0.769) BMI (0.053) DM (0.852) hypertension (0.133) NHT (0.446) PSA (<0.001) RDW-SD (0.035)	8
Wang et al., 2022 [18]	China	*Frontiers in Oncology*	retrospective, single-center	2018–2021	348	54	15.5	294	none	platelets (0.868)	6
Yamamoto et al., 2008 [19]	Japan	*International Journal of Urology*	retrospective, single-center	1994–2005	94	26	27.7	68	unequivocal presence of tumor cells in an endothelium lined space	cT stage (0.859) GS (0.053) PSA (0.022)	8
Yee et al., 2010 [46]	USA	*BJUI International*	prospective, single-center	2004–2007	1298	129	9.9	1169	unequivocal presence of tumor cells within an endothelium lined space	age (NA) PSA (<0.001)	8

AA = African American; BMI = body mass index; DM = diabetes mellitus; DRE = digital rectal examination; DSGb5 = disialosyl globopentaosylceramide; EGFR = epidermal growth factor receptor; GS = Gleason score; NA = not available; NOS = Newcastle Ottawa Scale; PNI = perineural invasion; PPBC = percent of positive biopsy cores; PSA = prostate-specific antigen; PSCA = prostate stem cell antigen; uPA = urokinase-type plasminogen activator; uPAR = urokinase-type plasminogen activator receptor.

**Table 2 ijms-25-00856-t002:** Quantity of articles analyzing clinicopathological prognostic factors found to be significantly correlated with LVI in at least two separate studies.

Predictor	Number of Studies	Patients with LVI/All Patients (%)
PSA	19	31,958/371,858 (8.6)
age	4	30,538/360,953 (8.5)
GS	4	695/5272 (13.2)
cT stage	4	241/2454 (9.8)
PPBC	3	121/1470 (8.2)
PSAD	2	83/649 (12.8)

LVI = lymphovascular invasion; PSA = prostate-specific antigen; GS = Gleason score; cT stage = clinical T stage; PPBC = percent of positive biopsy cores; PSAD = prostate-specific antigen density.

## Data Availability

The data supporting the findings of this systematic review and meta-analysis are available upon request. Please contact the corresponding author for access to the dataset.

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
