# Peer review of "Preoperative Factors for Lymphovascular Invasion in Prostate Cancer: A Systematic Review and Meta-Analysis"

_ijms, 2024, doi:10.3390/ijms25020856_

Round 1
Reviewer 1 Report
Comments and Suggestions for Authors
In this manuscript, the authors undertake a significant endeavor to identify clinicopathological factors that correlate with Lymphovascular Invasion (LVI) in prostate cancer. The study's strength lies in its comprehensive analysis, incorporating a vast pool of patient data to elucidate factors like preoperative PSA levels, clinical T stage, and biopsy Gleason score. These insights are crucial for the preoperative assessment and treatment planning for prostate cancer patients. A major concern is the omission of quality and risk of bias assessment in the manuscript (Line 80), which is crucial for systematic reviews and meta-analyses. The justification for this omission due to the inclusion of mixed methods studies is inadequate, as it is essential to evaluate the validity of the findings. This gap significantly undermines the manuscript's credibility, and the authors should incorporate an appropriate assessment tool to address this issue.
Line 23: The abstract clearly summarizes the key aspects of the study. Consider revising the conclusion to highlight the key implications of the findings or future research directions.
Line 52: The inclusion of a wide range of search terms is commendable; however, were there any specific search strings or combinations used?
Line 80: The authors state, "In terms of quality and risk of bias assessment, we decided to omit this step because most of the included studies employed both quantitative and qualitative methods. Therefore, an accurate assessment could not be obtained." This rationale is not acceptable in systematic reviews and meta-analyses, where the quality and risk of bias assessment are fundamental to ensure the validity of the findings. The use of mixed methods in the included studies does not preclude the possibility of performing such an assessment; instead, it necessitates a tailored approach to evaluating the quality and risk of bias that is appropriate for mixed-methods research. The authors should employ a suitable tool or framework that can accommodate the diversity of methodologies to provide readers with an understanding of the potential biases and the overall quality of the evidence base.
Suggestion: The authors need to justify the decision to omit the quality and risk of bias assessment more thoroughly, ideally by including it or explaining why it is not possible with the current data. They should explore established tools for quality assessment that can handle mixed-method studies, such as the Mixed Methods Appraisal Tool (MMAT), and apply it to their systematic review to enhance the manuscript's rigor.
Line 82: Were any assessments of heterogeneity, such as I^2 statistics or tests for publication bias, performed? Providing this information would enhance the robustness of the meta-analytical findings.
Line 97: While the PRISMA flowchart details the number of articles excluded at each stage, it would be informative to have a brief narrative description in the text accompanying the flowchart, summarizing the reasons for exclusions, especially the high number excluded for being not relevant to the review. Could the authors expand on what constituted relevance and non-relevance within the context of this review?
Other comments:
The manuscript could be strengthened by including a more nuanced discussion of the implications of age not being a risk factor for LVI. How does this align or conflict with current understanding or clinical assumptions?
Author Response
Reviewer 1 – authors’ responses
In this manuscript, the authors undertake a significant endeavor to identify clinicopathological factors that correlate with LVI in prostate cancer. The study's strength lies in its comprehensive analysis, incorporating a vast pool of patient data to elucidate factors like preoperative PSA levels, clinical T stage, and biopsy Gleason score. These insights are crucial for the preoperative assessment and treatment planning for prostate cancer patients. A major concern is the omission of quality and risk of bias assessment in the manuscript (Line 80), which is crucial for systematic reviews and meta-analyses. The justification for this omission due to the inclusion of mixed methods studies is inadequate, as it is essential to evaluate the validity of the findings. This gap significantly undermines the manuscript's credibility, and the authors should incorporate an appropriate assessment tool to address this issue.
Response: Thank you for your significant comment, which we considered crucial. After careful consideration of our methodology, we have opted to conduct a thorough reassessment of both the quality and risk of bias. The results are now presented in the article as Table S1 (supplementary – quality assessment using the Newcastle-Ottawa scale) and Figure 2 (risk of bias assessment). This adjustment stands as a pivotal step, enhancing the comprehensiveness of our article. Notably, the revelation that all included studies exhibited a high risk of bias holds paramount importance for readers, particularly in interpreting the results and conclusions.
Line 23: The abstract clearly summarizes the key aspects of the study. Consider revising the conclusion to highlight the key implications of the findings or future research directions.
Response: Thank you very much for your suggestion. We have incorporated an additional sentence at the conclusion of the abstract summarizing our study’s clinical implications and outlining directions for future research.
Line 52: The inclusion of a wide range of search terms is commendable; however, were there any specific search strings or combinations used?
Response: Thank you for your valuable suggestion. In section 2.1, we have included a comprehensive search strategy utilized across databases in our study.
Line 80: The authors state, "In terms of quality and risk of bias assessment, we decided to omit this step because most of the included studies employed both quantitative and qualitative methods. Therefore, an accurate assessment could not be obtained." This rationale is not acceptable in systematic reviews and meta-analyses, where the quality and risk of bias assessment are fundamental to ensure the validity of the findings. The use of mixed methods in the included studies does not preclude the possibility of performing such an assessment; instead, it necessitates a tailored approach to evaluating the quality and risk of bias that is appropriate for mixed-methods research. The authors should employ a suitable tool or framework that can accommodate the diversity of methodologies to provide readers with an understanding of the potential biases and the overall quality of the evidence base.
Suggestion: The authors need to justify the decision to omit the quality and risk of bias assessment more thoroughly, ideally by including it or explaining why it is not possible with the current data. They should explore established tools for quality assessment that can handle mixed-method studies, such as the Mixed Methods Appraisal Tool (MMAT), and apply it to their systematic review to enhance the manuscript's rigor.
Response: Once more, I appreciate your attention to this matter. We are confident that the tools employed will offer a transparent and comprehensible depiction of the risk of bias and the quality of the included articles. In terms of quality assessment, we decided to use Newcastle-Ottawa Scale. Your insightful suggestion on this aspect has been paramount to the refinement of this article, and for that, we express our sincere gratitude.
Line 82: Were any assessments of heterogeneity, such as I^2 statistics or tests for publication bias, performed? Providing this information would enhance the robustness of the meta-analytical findings.
Response: Thank you for bringing this to our attention. We acknowledge a nomenclature error that caused confusion. We have revised 'significant heterogeneity' to 'substantial diversity of data presentation.' It's essential to note that I^2 statistics, indicating substantial heterogeneity, were calculated, and are presented beneath each forest plot. In the mentioned sentence, our intent was to convey that certain articles were excluded from meta-analyses due to variations in data presentation. For example, in the case of PSA, 18 studies were excluded because they demonstrated an association with LVI in different ways, not conducive to meta-analysis methodology. To illustrate further, the study by Ferrari et al. (2004) highlighted a correlation between preoperative PSA and LVI but provided only the median PSA without mean, range, or IQR, preventing proper estimation for inclusion in the meta-analysis. Thus, it was not about data heterogeneity but rather about variations in data presentation. We appreciate your input, and hopefully, changes provided by us will clarify this distinction for future readers. In addition to changing this nomenclature error in subsections for every meta-analysis, we wrote a few sentences on it in the 2.4. Statistical analysis, to further point that it was not a biased decision, but decision based on lack of certain parameters in particular studies.
Line 97: While the PRISMA flowchart details the number of articles excluded at each stage, it would be informative to have a brief narrative description in the text accompanying the flowchart, summarizing the reasons for exclusions, especially the high number excluded for being not relevant to the review. Could the authors expand on what constituted relevance and non-relevance within the context of this review?
Response: Thank you very much for this suggestion. In section 3.1, we have included a brief description in the flowchart, providing further clarification on the criteria for studies categorized as 'not relevant to this review.' These criteria predominantly included studies on malignancies other than prostate cancer or those involving non-human research.
The manuscript could be strengthened by including a more nuanced discussion of the implications of age not being a risk factor for LVI. How does this align or conflict with current understanding or clinical assumptions?
Response: Thank you for your valuable suggestion. We appreciate your feedback, and in response, we have incorporated a more nuanced discussion on the implications of age not being a risk factor for LVI. This new section, located in the discussion, highlights the conflicting perspectives within the existing literature and underscores the challenges of synthesizing evidence from predominantly retrospective studies. We believe this addition enhances the depth of our analysis and provides a more comprehensive interpretation of our findings. Additionally, we provided appropriate citations.
Reviewer 2 Report
Comments and Suggestions for Authors
This article describes a systematic review and meta-analysis of studies investigating preoperative factors predictive of lymphovascular invasion (LVI) in prostate cancer (PCa). While an interesting topic with many identified relevant articles, the authors need to much improve the manuscript design and re-analyze all results in order to re-submit this manuscript for publication. For the below defined reasons, this manuscript is not ready or suitable for publication in its current state. Overall, the methodology is on the right track, but many crucial elements were left out or included without justification so that this work is extremely biased, unlike the goal of a meta-analysis, which is to provide an unbiased review and synthesize all existing data from all eligible studies into a single representative effect size. This has not been achieved, as the authors seemingly made unjustified decisions to include or exclude certain manuscripts from each stage of the analysis. In addition, either the authors are confusing the definitions for “qualitative” vs. “quantitative” data, there is some confusion as to the rules of a meta-analysis, which does not allow for mixing the two types of data, or the authors are not appropriately describing how they used or defined each respective type of data. This manuscript has the potential to be improved for publication at a later date, but many issues need to be addressed, such as inclusion of a description and analysis of heterogeneity, funnel plots for risk of publication bias assessment, tests for small-study effects if appropriate (e.g., Harbord’s), and even leave-one-out-meta-analyses for each listed outcome. See additional comments below:
This review is almost 1 year old based on the search date. It is recommended to update the search to reflect the current date.
Line 66: Define RP – it is assumed this is “radical prostatectomy”
Lines 79-81: If a risk of bias assessment was unable to be performed because results were both qualitative and quantitative, a meta-analysis may not be appropriate. The outcomes of interest for a meta-analysis must be consistent or in some way be converted from qualitative to quantitative values. In this case, a risk of bias could be performed and should be. Regardless, if a meta-analysis is performed, the bias assessment must be performed and reported.
Lines 82-88: It is unclear how a meta-analysis can be performed using distinct outcome variables (i.e., MD for continuous variables and OR for event rates for dichotomous variable).
Each section within the results states that “due to significant heterogeneity of the data…” It is unclear what about each study was heterogeneous and why it was listed as an included manuscript, but then excluded on the basis of being “heterogeneous.” Without a measurement of this, exclusion of some studies seems to create bias in the results. More explanation and justification need to be provided in order to remove studies from certain analyses. For example, what was the Cochran’s q test result for heterogeneity with leaving each combination of studies out of meta-analysis? What was the q cut-off level for determining whether to leave out certain studies? How was the combination of studies to leave out assessed? Not only does this completely invalidate the results of these meta-analyses to have such arbitrary inclusion criteria, but it completely invalidates the first section of results and renders Figure 1 completely useless.
Section 3.3 suggests that qualitative outcomes were only reported in the systematic review portion of this manuscript, but the question then remains as to why a risk of bias assessment was nor performed on the studies with quantitative data used in the meta-analyses. It is additionally unclear what defines a study as “qualitative,” as many of the referenced studies have quantitative outcomes. The limitation is that each study looked at one unique outcome not examined by other studies, but that is not what “qualitative” means.
Comments on the Quality of English LanguageMinor editing not influencing the overall understanding of the text -- proof-reading by a native English speaker may be helpful.
Author Response
Reviewer 2 – authors’ responses
This article describes a systematic review and meta-analysis of studies investigating preoperative factors predictive of lymphovascular invasion (LVI) in prostate cancer (PCa). While an interesting topic with many identified relevant articles, the authors need to much improve the manuscript design and re-analyze all results in order to re-submit this manuscript for publication. For the below defined reasons, this manuscript is not ready or suitable for publication in its current state. Overall, the methodology is on the right track, but many crucial elements were left out or included without justification so that this work is extremely biased, unlike the goal of a meta-analysis, which is to provide an unbiased review and synthesize all existing data from all eligible studies into a single representative effect size. This has not been achieved, as the authors seemingly made unjustified decisions to include or exclude certain manuscripts from each stage of the analysis. In addition, either the authors are confusing the definitions for “qualitative” vs. “quantitative” data, there is some confusion as to the rules of a meta-analysis, which does not allow for mixing the two types of data, or the authors are not appropriately describing how they used or defined each respective type of data. This manuscript has the potential to be improved for publication at a later date, but many issues need to be addressed, such as inclusion of a description and analysis of heterogeneity, funnel plots for risk of publication bias assessment, tests for small-study effects if appropriate (e.g., Harbord’s), and even leave-one-out-meta-analyses for each listed outcome.
Response: We sincerely appreciate your thorough review and the valuable insights you provided. Your comments have been instrumental in reshaping our manuscript. In response to your concerns, we have made significant improvements in our revised version:
- We have incorporated the Newcastle-Ottawa Scale for quality assessment (Table S1) and included a risk of bias assessment (Figure 2) to enhance the transparency and thoroughness of our evaluation.
- Additional details have been added to section 2.4 (Statistical Analysis), providing clarity on which studies were omitted during the meta-analysis, addressing concerns related to data inclusion and exclusion.
- We have made explicit in our revised manuscript that the actual heterogeneity of studies did not impact the selection of studies for specific meta-analyses. Every available study was included, and exclusions were solely due to the unavailability of crucial data on mean, range, and IQR, which prevented their conversion to means and standard deviations.
We believe these revisions address the concerns raised and contribute to a more robust and comprehensive manuscript. We greatly value your constructive feedback and are hopeful that our revised work aligns more closely with the standards expected for publication in your esteemed journal. Thank you for your time and consideration.
This review is almost 1 year old based on the search date. It is recommended to update the search to reflect the current date.
Response: Thank you for your thoughtful suggestion regarding updating our review. We appreciate your diligence in ensuring the currency of the information. However, at this stage of our study's progression, refreshing the data poses considerable challenges. The publication process in many journals often spans close to a year, making it challenging to incorporate the most recent studies. We acknowledge the importance of up-to-date information and assure you that we will consider this aspect carefully in future iterations of our research. Your understanding is highly appreciated.
Line 66: Define RP – it is assumed this is “radical prostatectomy”
Response: Thank you for this comment. ‘RP’ is defined as ‘radical prostatectomy’ in the introduction.
Lines 79-81: If a risk of bias assessment was unable to be performed because results were both qualitative and quantitative, a meta-analysis may not be appropriate. The outcomes of interest for a meta-analysis must be consistent or in some way be converted from qualitative to quantitative values. In this case, a risk of bias could be performed and should be. Regardless, if a meta-analysis is performed, the bias assessment must be performed and reported.
Response: We sincerely appreciate your insightful comment and have taken it into serious consideration. In response, we want to express that we have indeed performed a risk of bias assessment, as highlighted in our revised manuscript. Furthermore, we recognize that there may have been a misunderstanding in our use of the terms 'qualitative' and 'quantitative,' and we apologize for any confusion. We have clarified our approach in the revised manuscript, emphasizing that a risk of bias assessment was conducted appropriately. Thank you for guiding us towards a more accurate presentation of our methods.
Lines 82-88: It is unclear how a meta-analysis can be performed using distinct outcome variables (i.e., MD for continuous variables and OR for event rates for dichotomous variable).
Response: Thank you for your insightful comment. In response, we have incorporated additional details in section 2.4 (Statistical Analysis), specifically addressing our methodology concerning meta-analysis, including information on heterogeneity assessment and the consistent use of random-effects models. These additions aim to enhance the clarity and transparency of our statistical approach in the revised manuscript.
Each section within the results states that “due to significant heterogeneity of the data…” It is unclear what about each study was heterogeneous and why it was listed as an included manuscript, but then excluded on the basis of being “heterogeneous.” Without a measurement of this, exclusion of some studies seems to create bias in the results. More explanation and justification need to be provided in order to remove studies from certain analyses. For example, what was the Cochran’s q test result for heterogeneity with leaving each combination of studies out of meta-analysis? What was the q cut-off level for determining whether to leave out certain studies? How was the combination of studies to leave out assessed? Not only does this completely invalidate the results of these meta-analyses to have such arbitrary inclusion criteria, but it completely invalidates the first section of results and renders Figure 1 completely useless.
Response: Thank you for bringing this to our attention. We acknowledge a nomenclature error that caused confusion. We have revised 'significant heterogeneity' to 'substantial diversity of data presentation.' It's essential to note that I^2 statistics, indicating substantial heterogeneity, were calculated, and are presented beneath each forest plot. In the mentioned sentence, our intent was to convey that certain articles were excluded from meta-analyses due to variations in data presentation. For example, in the case of PSA, 18 studies were excluded because they demonstrated an association with LVI in different ways, not conducive to meta-analysis methodology. To illustrate further, the study by Ferrari et al. (2004) highlighted a correlation between preoperative PSA and LVI but provided only the median PSA without mean, range, or IQR, preventing proper estimation for inclusion in the meta-analysis. Thus, it was not about data heterogeneity but rather about variations in data presentation. We appreciate your input, and hopefully, changes provided by us will clarify this distinction for future readers.
Section 3.3 suggests that qualitative outcomes were only reported in the systematic review portion of this manuscript, but the question then remains as to why a risk of bias assessment was nor performed on the studies with quantitative data used in the meta-analyses. It is additionally unclear what defines a study as “qualitative,” as many of the referenced studies have quantitative outcomes. The limitation is that each study looked at one unique outcome not examined by other studies, but that is not what “qualitative” means.
Response: Thank you for your valuable suggestion. Indeed, it was another nomenclature oversight in our study, and we apologize for any confusion. We have rectified the term 'qualitative' to 'review' to signify that the presented data is not solely of a qualitative nature, as observed. Rather, the information below this sentence is the result of a systematic review and in-depth analysis of included studies. This clarification emphasizes that the data originates from studies that were not included in the meta-analyses we conducted. We appreciate your keen attention to detail.
Round 2
Reviewer 1 Report
Comments and Suggestions for Authors
Dear Authors,
Thank you for your thorough revisions in response to the comments and concerns that were raised. I am satisfied with the changes you've made to the manuscript. Based on your revisions, I will be recommending your paper for publication.
Reviewer 2 Report
Comments and Suggestions for Authors
The authors did a very good job in addressing the significant concerns with this manuscript.